# Signals of propaganda—Detecting and estimating political influences in information spread in social networks

**Alon Sela**[1,2]*, **Omer Neter**[3,4], **Václav Lohr**[5], **Petr Cihelka**[5], **Fan Wang**[3], **Moti Zwilling**[6], **John Phillip Sabou**[5], **Miloš Ulman**[5]

**1** Agricultural Engineering Department, The Volcani Agricultural Research Organization (ARO), Bet Dagan, Israel, **2** Department Industrial Engineering, Ariel University, Ariel, Israel, **3** Department of Computer Science, Bar Ilan University, Tel Aviv, Israel, **4** Microsoft Security Research, R&D Center, Herzeliya, Israel, **5** Department of Information Technologies, Faculty of Economics and Management, Czech University of Life Sciences Prague, Prague, Czech Republic, **6** Department of Economics and Business Administration, Ariel University, Ariel, Israel

* alonsela@volcani.agri.gov.il

**Data Availability Statement:** All the code is available in the Github repository of the study: https://zenodo.org/records/10805274 or https://github.com/OmerNeter/tweets_politics_project

## Abstract

Social networks are a battlefield for political propaganda. Protected by the anonymity of the internet, political actors use computational propaganda to influence the masses. Their methods include the use of synchronized or individual bots, multiple accounts operated by one social media management tool, or different manipulations of search engines and social network algorithms, all aiming to promote their ideology. While computational propaganda influences modern society, it is hard to measure or detect it. Furthermore, with the recent exponential growth in large language models (L.L.M), and the growing concerns about information overload, which makes the alternative truth spheres more noisy than ever before, the complexity and magnitude of computational propaganda is also expected to increase, making their detection even harder. Propaganda in social networks is disguised as legitimate news sent from authentic users. It smartly blended real users with fake accounts. We seek here to detect efforts to manipulate the spread of information in social networks, by one of the fundamental macro-scale properties of rhetoric—*repetitiveness*. We use 16 data sets of a total size of 13 GB, 10 related to political topics and 6 related to non-political ones (large-scale disasters), each ranging from tens of thousands to a few million of tweets. We compare them and identify statistical and network properties that distinguish between these two types of information cascades. These features are based on both the repetition distribution of hashtags and the mentions of users, as well as the network structure. Together, they enable us to distinguish ($p-value = 0.0001$) between the two different classes of information cascades. In addition to constructing a bipartite graph connecting words and tweets to each cascade, we develop a quantitative measure and show how it can be used to distinguish between political and non-political discussions. Our method is indifferent to the cascade's country of origin, language, or cultural background since it is only based on the statistical properties of repetitiveness and the word appearance in tweets bipartite network structures.

Data cannot be fully shared publicly because of its size and because we are not its owners. Due to the sizes of the datasets, which are over 200 MB (even when zipped), and also since we are not the owner of the data but rather downloaded it from open repositories, we added links to the data sets locations in the project's Git repositories. We also added several datasets to enable a smooth run of the code. The additional datasets are found in links in the repository. These are links to (1) Kaggle, (2) Fishgar and (3) the Digital Library data repository. In the root of the Git repository, we added a table named "DataSources.xlsx" where we specify for each data source the link for downloading it. Additionally, statistical properties for each of the network data sources (#clusters, #nodes, #edges, Slope) is found in the "Network_dataset.csv" file in the root of the repository. The minimal data set underlying the results described in your manuscript can be found in the GitHub repository, https://github.com/OmerNeter/tweets_politics_project in files 'Fig 4 A-D.xlsx' and 'Fig 5.xlsx'.

**Funding:** The fund support was financial. The Ariel Cyber Innovation Center in conjunction with the Israel National Cyber Directorate in the Prime Minister's Office; Award Number: None | Recipient: Alon Sela, Ph.D. This grant was used to pay for the programmer of this project, Mr. Omer Neter which is also one of the authors of this paper, and we declare that the funders had no role in study design, data collection and analysis, decision to publish, or preparation of the manuscript or any other scientific or non scientific matter other than providing the initial funds.

**Competing interests:** The authors have declared that no competing interests exist.

## Introduction

In most democratic systems, politicians need to convince the masses to choose them over other candidates. One important technique that is commonly used in political struggles includes repetitive broadcasting [1, 2] of a simple and clear message. These focused and short messages, also called slogans [3], tend to be catchy, simple, and clear. Examples of such political slogans are *"Make America Great Again"* used by Donald Trump in the US Presidential Election of 2016, or *"Yes We Can"* used by the Obama 2008 campaign. Such slogans in political campaigns have a greater degree of repetitiveness compared to "normal" non-political discussions.

Pre-election political messages tend to be more aggressive [4], populistic [5] and in general, use similar techniques as the ones used in commercial communication [6, 7]. These commercial communication techniques repetitively broadcast a few well-defined messages, to penetrate the minds of the exposed audience and affect the memory recall for a product [8] or similarly, a political player.

Political campaigns use massive broadcasts to spread their messages and capture the voter's attention. Such mass broadcasts are generally performed in many mediums in parallel and in high volumes. They have become an essential technique in modern Russian propaganda. Such propaganda methods use *"rapid, continuous, and repetitive"* broadcast, in a *"high numbers of channels"*, and have *"a shameless willingness to disseminate partial truths or outright fictions"* [9]. While such propaganda techniques, which are sometimes referred to as "Brainwashing" are associated with Communist or Fascist regimes, it is naive to assume they are not applied in modern Western democracies.

For example, massive use of bots to increase the repetitiveness of a message has been detected in Canada [10], India [11], the UK [12], the US [13] elections and are probably operating in any country where elections occur. Bots are complex automated accounts that operate in a social network and help spread their master's messages. Bots include a wide range of NLP techniques, e.g., they use fake accounts, cyborgs (human-machine cooperation), and search engine manipulations [10, 12–16]. Furthermore, with the latest developments in large language model techniques, and the effort to shrink these models' computational resources such that they can be used by everyone, bots are expected to become even more sophisticated and harder to detect.

In addition to bots (of which some are illegal since they pretend to be real humans), information spreaders also use legal methods to increase the spread of a political agenda and increase one's influence including the use of opinion leaders, or social media management platforms [14, 17]. One should note however that we use in this study the term "propaganda" in the context of an effort to change the public's opinion. While the initial use of this term has been mainly associated with totalitarian regimes, it has also been used for many years, in softer regions of marketing [18]. We use this term in its wider and contemporary context [19], which defines propaganda as tools that include *"filtered digital content, targeted advertising, and differential product pricing to online users"*. We follow this line and consider propaganda as any intended effort to push a message or an agenda, to as large as possible audiences, as opposed to simply publishing the message on the web and having others comment and discuss the content of the message through a natural discussion. This wide definition enables us to move from the negative association of "bad Russia vs. good USA" to a more quantitative and neutral definition which includes any intervention in the natural process of information spread. In this aspect, any centralized (or decentralized) effort to change the opinion of the masses is to some degree, seen as propaganda, even when the causes are good and true.

As such, the use of bots, either by a classical Russian information bureau or by a commercial communication firm is considered here as some type of propaganda. For a broader review

of the topic of bots, cyborgs, and other modern information manipulations, their detection methods, their working mechanism, and their goals, we recommend the broad taxonomy of [15].

Social network platforms such as Twitter (now named "X"), YouTube, and Facebook, constantly struggle against powerful and sophisticated entities that try to manipulate their platforms. These manipulation either by bots or by other algorithmic methods, operate on behalf of a country, state-sponsored agencies, individual politicians or stock manipulators, or simply people aiming to spread conspiracy theories. All these parties are constantly trying to change and influence the opinions in their society, by mass broadcasting includes an effort to change and influence the opinions in society by massive broadcasting of specific agendas on social platforms.

In the current study, we compare 22 data sets of large information cascades. These include 10 cascades related to political topics and 12 cascades related to non-political ones. We use as a comparison group for the **political cascades**, i.e., the information cascades that were related to the political topics, several non-political data sets related to disasters. We believe that disaster-related information cascades include lower direct gains from the manipulation of information and are therefore probably a good candidate for a comparison group to the political cascades. This belief is based on the following arguments:

1. Natural disaster-related information cascades might be less biased because a disaster or national emergency is experienced by larger portions of the society (less likely to be extremist or marginal parts of society). This applies even if the disaster is regional, e.g., the 9/11 terror attacks, the 2011 Fukushima nuclear disaster, the 2013 Vanuatu tsunami, the 2015 Nepal earthquake, the 2017 Gulf Coast hurricanes, etc. In these cases, many of the users expressing their opinions are "regular users" with no aim nor experience in methods to spread their message. They communicate on social networks simply to share, get informed, and help others.

2. People's tendency to share their experiences in hard times is natural. It is not for the sake of financial or power gains, but rather as a natural humanistic act higher [20–22].

3. Disasters cascades are less likely to be manipulated because there is less to gain from such events. Also, in the early stages of a disaster, its final consequences are still unclear. While politicians might relate themselves to good management of a catastrophic event, they will probably be more careful not to be perceived as opportunistically promoting themselves on the backs of their suffering people.

4. Overall, events of disaster seem to contain a genuine human need of "regular people" to share and seek information. Political discussions on the other side are aimed at convincing others of the rightness of the ideologies of one party and the falseness of the ideology of the opposing party.

**We, therefore, hypothesize that political cascades, as opposed to disaster cascades, will include higher levels of repetitiveness due to their rhetoric nature**. By comparing the statistical properties of the politically-related cascades to those related to disasters, we hope to differentiate between these two types of information cascade, thus possibly revealing large-scale signals of external interventions/information manipulation, which we name: **"Signals of Propaganda"**.

The rest of the article is ordered as follows. In the Background section, we introduce some of the most relevant literature related to "classical" and computational propaganda. We also explain some essential theories related to power law distribution, which is the base core for our statistical analysis. The Material and Methods section then describes the data sets, the data

collection process, and the methods used to study the power law distribution [23, 24]. We also show additional separating properties of the two types of cascades. The code used to conduct the main analysis of this part can be found in the repository (DOI 10.5281/zendo.10805274) of the work. We also show additional separating properties of the two types of cascades based on their transformation to a bipartite graph connecting words and tweets and develop a quantization measure to detect bias. The Results section presents a comparative visualization of the two types of cascades based on these separative features. It also presents the separation by the mathematical quantity that we develop to distinguish between political and nonpolitical cascades. The Discussion and Conclusion sections finalize the study, address some known limitations, and propose possible future research directions.

## Background

### Historical overview of propaganda

Propaganda is an inseparable part of the history of wars. It was used in ancient times as a psychological method to weaken one's enemies, by spreading blends of scary stories with mystic, unrealistic, or grandiose events, stories that strengthen one king's image while weakening the perceived power of the opponent. For example, Roman troops were known for their brutal ways of planting fear in their opponents' hearts. These methods helped them defeat rebellious populations as horror stories of their cruel war acts moved faster than the troops themselves [25]. In the early 20[th] century, propaganda was redefined again [26]. Both the Nazi fascist propaganda machine and the Communist Cold War propaganda methods used mass media to mold collective beliefs and control their citizens. The most important difference between the 20[th] and the 21[st] century propaganda is the medium in use. The Cold War propaganda mainly used mass media (radio, newspaper, or TV) as its medium for spreading its ideology. Unlike mass media that can be controlled by governments, social networks, especially when consumed via smartphone applications, are based on a many-to-many communication channel. This forms a different spreading dynamic that requires the multiplication of social network accounts, i.e., the use of fake personas, to spread messages by governmental agencies and/or private parties alike, and also, to trick and manipulate the recommendation agents (search engines) [27] that connect news to users consuming this news.

### Propaganda and the repetition of messages

The relations between the degree of repetition of a message and its effect on the person receiving the message are complex. The experimental work of [8] has found that repetition first increases the rates of recall from one's memory, but then, with more than 4 or 5 exposures, further repetitions decrease the recall. Other studies found similar results but also claimed that the persuasiveness levels of the message mediate this effect [28]. In general, politicians tend to use higher than average levels of repetition in their rhetoric speeches [29]. The Nazi Propaganda Minister Josef Goebbels, one of the most notorious propaganda masters, summarized his principles of propaganda in his diary [30] where he claimed that propaganda *"must be utilized again and again"*, and also that *"A propaganda theme must be repeated, but not beyond some point of diminishing effectiveness"*. Also, that *"propaganda must label events and people with distinctive phrases or slogans"*.

### Computational propaganda

Modern propaganda is sometimes known as computational propaganda [12, 13, 31]. This new type of propaganda abuses the same old sociological and psychological patterns as the 20[th]

century propaganda while using new technologies and tools. It is recognized by repetitiveness, a blend of true and false messages, and the use of fear and anger to help spread the messages. Computational propaganda methods include, for example, organized groups that disseminate similar messages into online social media platforms [32] and the synchronization of these groups to increase their web presence. Also, techniques such as an early "ping-pong" like the exchange of messages within *Spreading Groups* [14] enhance the future spread of a message and propagate it to larger audiences.

The operation of groups of bots blended with real users, (also named "cyborgs"), is another method to trick the social network algorithm [33]. Twitter and Facebook publish officially their efforts to remove such groups of bots and fake accounts from their platforms, but also, in parallel, permit the creation of automated programmable APIs [34] that enable the construction of such tools. Regardless of the debate if the media giants can or cannot stop the use of bots, it is clear that the number of bots continuously grows [35]. Bots were found to influence public opinions in Chinese political cascades on Twitter and Weibo. Interestingly, not only anti-governmental messages use bots, but also pro-government actors [36]. Bots were found to operate in the UK pre-Brexit debate [12] and are used by countries as well as by private entities that operate groups of bots, cyborgs, and trolls [31] to spread their messages.

While most researchers agree that bots operate in most social network platforms, their identification seems to become harder. The recent advancements in large language models [37, 38] such as GPT-4–5 with the ability of AI to smartly imitate human patterns [39], increase the sophistication of bots operations in social platforms. The sophistication of bots is growing and is expected to further grow. Also, in the period of Information Overload [40] we are likely to see an increasing noise-to-signal ratio between alternative and real truth, making the detection of the former harder. While bots and similar professional information spreaders detection is likely to become harder than ever before, the goal of bots is kept unchanged. Their goal is to increase the size of their audiences as they spread info-bites to us—their human customers. Thus, we need a method to capture their influence through the statistical properties of manipulation of information spread. These directions will be explained in the next section.

## Long tailed distribution in information cascades

The normal distribution is the most fundamental in statistics and is applied in almost every scientific discipline. Its importance is derived from the Central Limit Theorem and the fact that the sum (or average) of a large enough number of reoccurring experiments, e.g., independent coin flips with a probability $p$ of success is distributed according to the normal distribution. This robust statistical tool has an important underlying assumption that needs to be fulfilled—that the results of each experiment are not dependent on previous results. When a researcher inspects the effect of a given drug on any defined disease, the researcher first needs to verify that there is no interaction between the different subjects of the experiment where the success rates of one trial will influence the outcome of the other.

In some cases, however, this assumption does not hold. If the researcher studies the sizes of cities, the growth of one city might not be independent of its current size. A large city attracts more people than a small town, thus city growth is not independent, but rather dependent on its size. They will thus be distributed according to a power law distribution [41] and not a normal distribution as would be expected in the case of independent trials.

This idea is critical for the analysis of information cascades. Power law distributions are found in a different class of phenomena where positive feedback exists. Not only city populations and sizes of cities, the intensity of wars, the sizes of electrical blackouts, the relations between palm heights and diameter [42] or the number of citations of academic articles, all

have positive feedback in their growth dynamics and as can be easily seen, are all distributed according to power law (or long-tailed) distributions [41]. While there are claims that fitting data to Power Law by graphical methods based on linear fit on the log-log scale is biased and inaccurate [43], other researchers claim that binning is a good method, and that the critics on the LME methods have no real base [44]. We thus applied an exponential binning on the data, which helps removing the possible bias due to the long tail of the distribution. In regards to information cascades, such positive feedback exists in modern information cascades, since the "hotter" a topic is, the more people discuss it. The more it is discussed, the more it captures people's attention, which positively influences the cascades' dimensions. We thus expect information cascades to follow a power law distribution.

The relationship between Power Law and propaganda passes through the positive feedback mechanism that in one of the processes that create power law. Propaganda is not effective if it does not include the echoing of the message through the authentic audience. We showed in a previous work on this topic [14], that an effective way to spread a message is by echoing it first in a recruited group, which we name "spreading group", and then from this echoing, the message spreads outside the group to the authentic users. We also name this group of initial spreaders also "message detonator" since they act to activate the spread. This process, where authentic users are more likely to believe to a message when they are exposed to it from several directions thus tend to spread it themselves, is exactly the positive feedback mechanism that we believe creates the power law distribution. Furthermore, the data simply shows that the hashtags and users are distributed according to a long tail distribution. This usually implies some dependence or positive feedback in the process, compared to a sum of independent Bernoulli processes which result in a Normal distribution.

## Materials and methods

### The data

We used three different data repositories and over 13 GB of data to analyze the difference between the political and disaster cascades. First, we downloaded 6 data sets related to political information cascades from open source repositories such as Kaggle [45], and 6 other datasets related to disasters from the Digital Library Repository [46]. The sizes of these data sets are shown in S1 and S2 Tables in the SI section. To ensure the collection and analysis method complied with the terms and conditions for the source of the data. and also due to the sizes of the datasets, we published in the repository of this article the links to all datasets, and only included a small data set to demonstrate the code itself. All code, links to the open-source datasets and the tables used in this article are found at the repository (DOI 10.5281/zendo. 10805274) of the work.

These data sets are used to compute the slopes of the distributions, reflect the speed of decay in each field, and compare these macro-scale properties of political and disaster events. The distribution of users/hashtags on a log-log scale and the computed exponent slopes of these datasets distributions are presented in Figs 2 and 3 respectively. The data was binned on a logarithmic scale before the slope estimation to eliminate the long-tail bias—a common practice in this field. we further Elucidate in the discussion section for the reasons of using this method. The slopes in power law distributions represent the exponent of the distribution $Y \sim c \cdot x^{-\alpha}$, where $\alpha$ is the exponent determining its speed of decay.

For constructing the bipartite graphs, we used the data sets from Kaggle for the political cascades, and for the disasters, we used a third data repository—the Figshare [47]. Here, since we could not get initially the entire tweet message, which was required to create the word-to-tweet graph, but only the tweet ID. We thus used a Twitter scraping tool named Hydrator [48, 49],

which scrapes Twitter and collects tweet messages according to tweet ID. to use this tool, one first needs to subscribe to the Twitter (now X) developer platform, and receive appropriate keys and tokens from the Twitter platform API. We use this tool after verifying that its use fully complies with the terms and conditions of Twitter, as clearly defined in their developer platform [50]. The full list of datasets used to compare the word occurrence on the bipartite graphs is presented in S3 Table in the SI section.

## Quantifying the bias in information cascade

Based on the findings described above, we continued and developed a quantitative measure to differentiate between political and non-political cascades. Note that the measure is language-independent since it is based on the internal statistical property of propaganda to include higher repetition rates.

To explain the quantitative measure, one needs to note that the main difference between the distributions of hashtags and the curve-fit line, is the deviation of some points from the line toward the upper left side. This deviation can be observed in the political cascade (left column) of Fig 2. The deviation reflects several hashtags in the political discussion which repeat more than would be expected.

Based on this pattern, we develop a new measure that captures the political bias, named *Power law MSE (PLMSE)*. The name was given since its development is based on the concept of mean square error (MSE) with an adaptation to fit power law distributions, i.e. PL MSE.

The measure is presented in Eq 1 below, where $n$ is the number of points in the histogram of the cascade, $i$ defines the index of each point when the data is sorted from most frequent to least frequent, $y_i$ is the frequencies of a hashtag in index $i$ when sorted from the most common to the least common, for any observed value at $x = i$, $\alpha$ represents the slope of the least square optimal curve-fit line and $c$ represents the intercept of this linear line on a log-log scale. Note that the first term of the equation $\frac{1}{i \cdot (i+1)}$ is the weight of each point in the final PLMSE score. The first points, i.e., $i = 1$ receive the highest weight. We further discuss this weighting method in the discussion section. The second term in the summation; $\log^2\left(\frac{y_i}{c}\right)$, is the square error of each point, i.e., the distance of each point from the theoretical power law distribution curve on a log scale, where $i$ is the point on the x-axis, $\alpha$ is the power law exponent (which on a log-log scale appears as a simple slope), and $c$ is the intercept. Last, the term $\frac{n+1}{n}$ ensures that the value of all the weights in the PLMSE score will always be equal (regardless of the number of points $n$) to 1.

$$PLMSE = \frac{n+1}{n} \cdot \sum_{i=1}^{n} \frac{1}{i \cdot (i+1)} \cdot \log^2\left(\frac{y_i}{c} \cdot i^{\alpha}\right) \qquad (1)$$

## Explaining the PLMSE formula

The following section explains the PLMSE formula and each of its components. Overall, Eq 1 has 3 main components: (1) $\frac{n+1}{n}$, (2) $\frac{1}{i \cdot (i+1)}$ and (3) $\log^2\left(\frac{y_i}{c} \cdot i^{\alpha}\right)$. Let us start from part (3). We denote $i$ as the index of the bin (x-axis) and similarly $y_i$ is the frequency of the distribution at point $i$, in our case the y-axis. To compute a simple MSE, on a linear line, we need to compute the expression $\frac{1}{n} \cdot \sum_{x=1}^{n} \left(y_i - (a \cdot i + b)\right)^2$ where $a$ and $b$ are the linear line coefficients. In our case, where our data is not linear, assuming it is a power law distribution, then it should be linear on a log-log scale.

We thus transform:

$$\hat{i} : x \rightarrow log(i)$$

$$\hat{y}_i : y_i \rightarrow log(y_i)$$

$$line : y'_i = -\alpha \cdot \hat{i} + log(c)$$

We now define the squared error in terms of the Log-Log line:

$$\sum_{i=1}^{n} w_i \cdot (\hat{y}_i - y'_i)^2 \rightarrow \sum_{i=1}^{n} w_i \cdot (log(y_i) - (-\alpha \cdot log(i) + log(c)))^2$$

$$\sum_{i=1}^{n} w_i \cdot (log(y_i) - (-\alpha \cdot log(i) + log(c)))^2$$

$$= \sum_{i=1}^{n} w_i \cdot (log(y_i) - log(c) + (log(i^\alpha))^2$$

$$= \sum_{i=1}^{n} w_i \cdot \left(log\left(\tfrac{y_i}{c}\right) + (log(i^\alpha)\right)^2 = \sum_{i=1}^{n} w_i \cdot log^2\left(\tfrac{y_i}{c} \cdot i^\alpha\right)$$

As for part (2) of the equation, we constructed modified weights that give a higher focus (weight) on the upper left data points in the straight line. This uneven weighting system is needed since in the power law distribution, these points represent many of the observations, while the lower right parts of the distribution (the long tail) represent only a few points. Thus, to give a fair representation as required to compute the slope properly, and to construct proper weights, first notice that the following equation holds:

$$\sum_{i=1}^{n} \frac{1}{i \cdot (i+1)} = \sum_{i=1}^{n} \left(\frac{1}{i} - \frac{1}{i+1}\right) = \left(\frac{1}{1} - \frac{1}{2}\right) + \left(\frac{1}{2} - \frac{1}{3}\right) + \ldots + \left(\frac{1}{n} - \frac{1}{n+1}\right) =$$

$$1 - \frac{1}{n+1} = \frac{n}{n+1}.$$

Then, as for part (1) of the equation, we need this part such that the normalized weights in part (1) and their multiplication by part (2) of the PLMSE equation, are **always** equal to 1.

$$\sum_{i=1}^{n} w_i = \frac{n+1}{n} \cdot \sum_{i=1}^{n} \frac{1}{i \cdot (i+1)} = 1$$

## Bipartite graphs transformations

The comparison between the political and non-political cascades based on hashtag distribution alone can suffer from an internal bias. The reason for this is a possibility that political users might tend to use hashtags more often than regular users. To correct this issue, we want to look at the words used in the tweet regardless if they are hashtags or not. Thus, we transformed the data and constructed a bipartite graph (see illustration in Fig 1). In this graph, we can compare the word recurrence in the tweets themselves and bypass the possible problem of uneven use of hashtags in political and non-political actors.

Started by a conventional stemming and lemmatization process that removed common words such as "and / "or" / "if", etc. from the tweet text. These steps were performed by the NLTK python package. Then, we constructed a link between tweets and words appearing in these tweets connecting words appearing in each tweet to create a weighted link between words and their tweets for each tweet, where the weight is according to the number of times

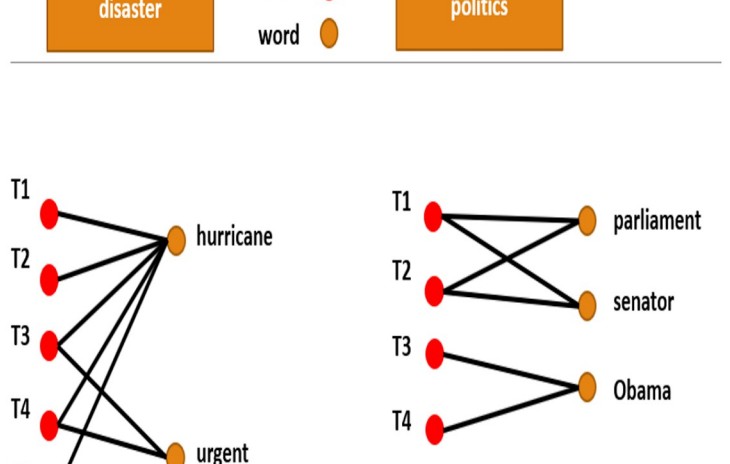

**Fig 1. Illustration of word-to-tweet bipartite graph.**

similar words appear together. Fig 1) illustrates this process. These links form a graph for each cascade, connecting words to tweets. These bipartite graphs were constructed both for the political and for the non-political topics. We computed and compared several graph properties on these bipartite graphs, e.g., degree distribution, number of communities, and average degree. We also looked at network and node properties such as the betweenness centrality, the variance of the node's degree, the average largest component size (GC), and the distribution of component sizes. Last, we used the properties that were found most significant in the bipartite graphs as a separative construct.

## Results

We show first the slopes of each of the 12 distributions—for the 6 political cascades and the 6 disasters cascades. We use the Least Square Method after binning the data on a logarithmic scale [44] to find the slopes on a log-log scale, and comparing the exponent parameter $\alpha$ of the power law distribution $y(x) = c^{-\alpha x}$ between the two groups.

We see a clear difference between both the distribution of hashtag repetitions and the distribution of user repetition between the political and the disaster cascades.

The smaller mean slope in the hashtags' appearance for the political cascades reflects a slower decay of hashtag frequency. This is due to some hashtags that repeatedly appear in the political messages (repeating hashtags in political slogans), while for the disaster messages, there is only one such hashtag, i.e., the name of the disaster itself. After inspecting the hashtags more deeply, we find that in the disasters, the name of the disaster event itself, e.g. *#florence*, *#dorian*, *#harvey*, etc. appears in almost all the tweets. In contrast, in the political cascades, we observe (left column of Fig 2) several points above the line reflecting hashtags that appear repeatedly more frequently. Examples of political terms that were commonly repeated in the

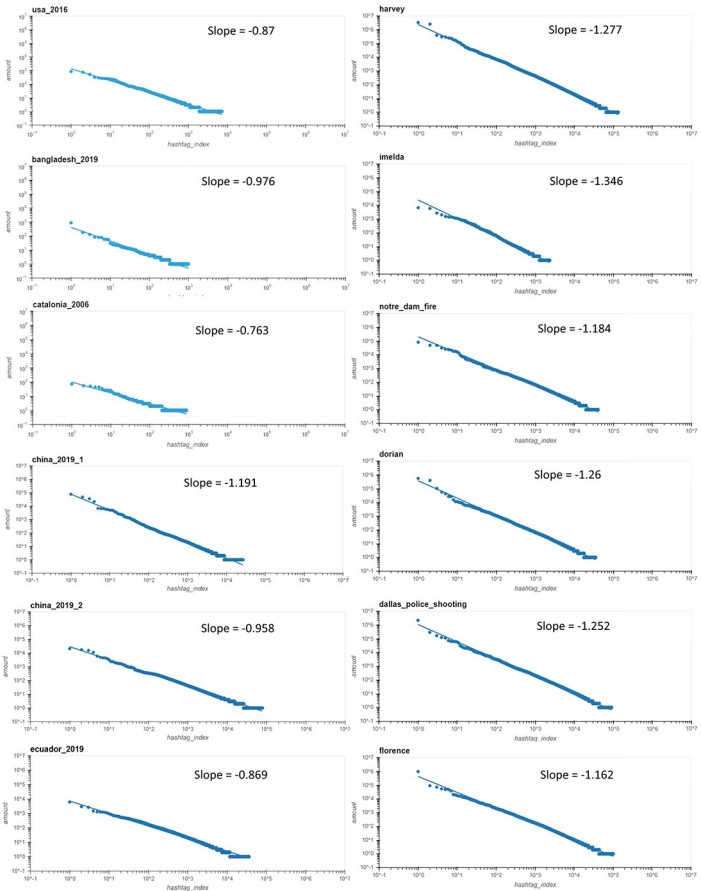

**Fig 2. Distribution of hashtags on a log-log scale for 6 political (left column) vs. 6 disasters (right column) cascades.** We only show here 6 disaster and political cascade distributions, while the scores and PLMSE scores for all the data are shown in Fig 4(C) and 4(D). Mean slopes are -0.938 / -1.214 for the political/disasters cascades, a significant difference (t-test, p-value = 0.0041). For the political cascades, most slopes values $\alpha < 1$, while for the disasters cascades, all $\alpha > 1$. When using all 10 political cascades (and not only 6 cascades) the significance even grows (slopes = -0.973 / -1.208, t-test, p-value = 0.00187).

political cascades are *"#MakeAmericaGreatAgain", "#ImWithHer", "#HillaryClinton", "#NeverHillary", "#realdonaldtrump","#NeverTrump" or "#fakenews"* in the 2016 USA elections. These demonstrate the repetitiveness nature of political slogans.

### Slopes (exponents) of hashtags distribution

One important and meaningful property in power law distributions is their slopes. Note that a power law graph represents a general function of a type $y = c^{-\alpha x}$, where c is a constant and $\alpha$ is the exponent (the slope when on a log-log plot). We can observe in Fig 2 that the slopes are generally smaller in the political cascades (left column) compared to the non-political ones (right column). The slopes of the Hashtags distributions in the political cascades (left column of Fig 2) are in the range $\alpha = 0.763$ for the Catalonia politics cascade and $\alpha = 0.976$ for the Bangladesh politics cascade. When compared to the disaster cascades, we can see that most cascades have steeper slopes (i.e. a larger absolute value). The average slopes for the political cascades are $\bar{\alpha} = 0.97$ while for the disasters cascades, they are $\bar{\alpha} = 1.21$. These values differ significantly (t-test, p-value = 0.0018). The sharper slopes in the power law distribution of the

disaster cascades (i.e., larger absolute values of the negative slope), represent a sharper decline in the likelihood of a hashtag to repeat. This can result from many messages and hashtags that only appear once or twice, and very few hashtags that appear more times. In comparison, in the political cascades, a larger number of hashtags appear repeatedly in many of the political campaigns, as the political players use rhetorical repetition in their messages to implant the messages in their audience's minds. Note that in the power law distribution equation, the probability $p$ of an event occurring $x$ times; $p(X = x) = c \cdot x^{\alpha}$ where $\alpha$ is the slope, and $c$ is the y-intercept. Thus a larger (absolute value) of $\alpha$ implies a faster decay of the power law slope and a sharper difference between hashtags appearing many times and those appearing only a few times. We can see that the deviation from the slope in the upper left side of the curve is a result of these several hashtags that appear in the political cascades more than they should appear naturally, suggesting higher repetition levels than we would expect. More importantly, the difference between the slopes of the two types of cascades is significant (t-test, p-value = 0.002) when accounting for all 10 political cascades compared to the 6 disaster cascades respectively as seen in Fig 4, and also it is significant when only using the 6 political cascades as in Fig 2 (t-test, p-value = 0.004). These results suggest a clear difference between the hashtags distributions of the political and disaster cascades.

## Slope comparison for user's distributions

A comparison of the user's distribution slopes and scores for the disasters and the political cascade groups is presented in Fig 3. The left column presents the political cascades while the right column presents the disasters cascade. The user distribution differs substantially between the two topics. In disasters, the points fit rather well the theoretical power law curve. This, however, is not the case in the political cascades group (left column), where we observe a sharp decrease in the lower right part of the distribution. We are not certain of the reasons for this sharp fall.

In the disaster cascades, the slopes are substantially lower. For the users distributions of the political topics, the exponents slopes are in the range between −1.32 and −3.09 compared to the disasters discussion where the exponents slopes were ranging between −0.545 and −0.664. These differences are clearly significance (t-test, p-value = 1.4E-05). The sharper decay in the political users can be understood as the strong emotional aspect of political discussions, where *—one can either be talking about politics, one or does barely talk about politics at all* [51]. In the disaster cascades, the distribution of user engagement is more evenly distributed and better fits the "natural" power law distribution curve which is expected.

## Separation through word-to-tweet bipartite networks

In addition to the method described above of separation between political and non-political cascades based on the exponent (slope), we also transformed the data into a bipartite network. We show that this transformation improved the separation. Furthermore, we use our newly developed quantitative measure—the PLMSE in Eq 1 and show how it can be used to better separate between political and non-political cascades.

This separation is observed in Fig 4 (without the network transformation) and with the network transformation in Fig 5 below. First, in Fig 4, we show the simpler separation by the user's slopes (A) and the user's scores (B) where (A) separates the cascades by the slope while (B) by the PLMSE scores. In both sub-images, the orange points are the political cascades while the blue points are the disaster cascades. In (C) and (D), we show a similar separation, but now by the slopes of the distributions of hashtags (C) along their PLMSE scored (D).

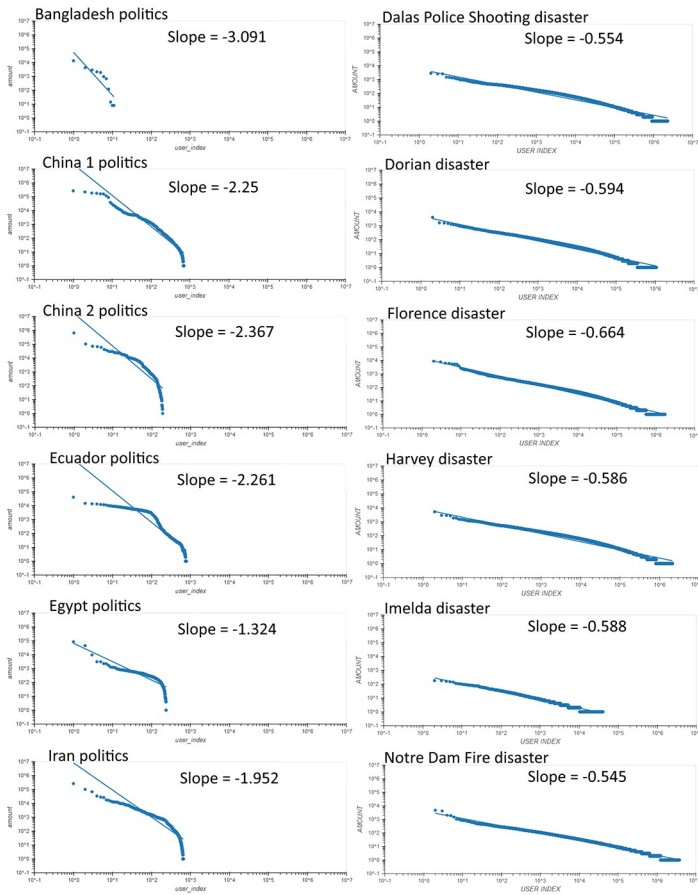

**Fig 3. Distribution of users on a log-log scale for political (left column) vs. disaster (right column) topics.** As in Fig 2, we show here 6 disasters compared to 6 political cascades. The mean slopes of political topics are -2.21 compared to -0.589 for the disaster cascades, a clearly significant difference (t-test, p-value = 1.4E-5). Full slopes and scores comparisons when accounting for the additional 4 political cascades are present in Fig 4(A) and 4(B).

We can observe a good separation between the political and the disaster cascades in (A) and (B), along a different mean but without a single line separation in the Hashtags slopes (C) and their scores (D) where the means differ but not all points.

In the distribution of hashtags (C), the slopes in the political topics (orange) (blue) are generally smaller compared to the disasters. One should note that a small slope indicates a slower decay, and thus more words and hashtags that appear many times compared to the disasters, where hashtags appearing many times are less frequent. This is possibly due to the more repetitive use of words (and hashtags) in the political cascades, resulting from the rhetoric propaganda and a style that uses repetitions, i.e., brainwashing of slogans. Also, since the users' slopes alone do not separate between the two types of cascades well enough, we add to the separation the bipartite graphs transformation.

In Fig 5, we show the political / disaster separation on a two-dimensional space, (A) and (B) show only the s of degree distribution slope (A) and the average degree (B) in the bipartite network. Images (C) and (D) add these dimensions to the number of clusters in the bipartite network.

To construct a bipartite graph from each cascade, we used the entire tweet message, which was not available in the initial datasets. We, therefore, use another set to collect 6 disaster-related

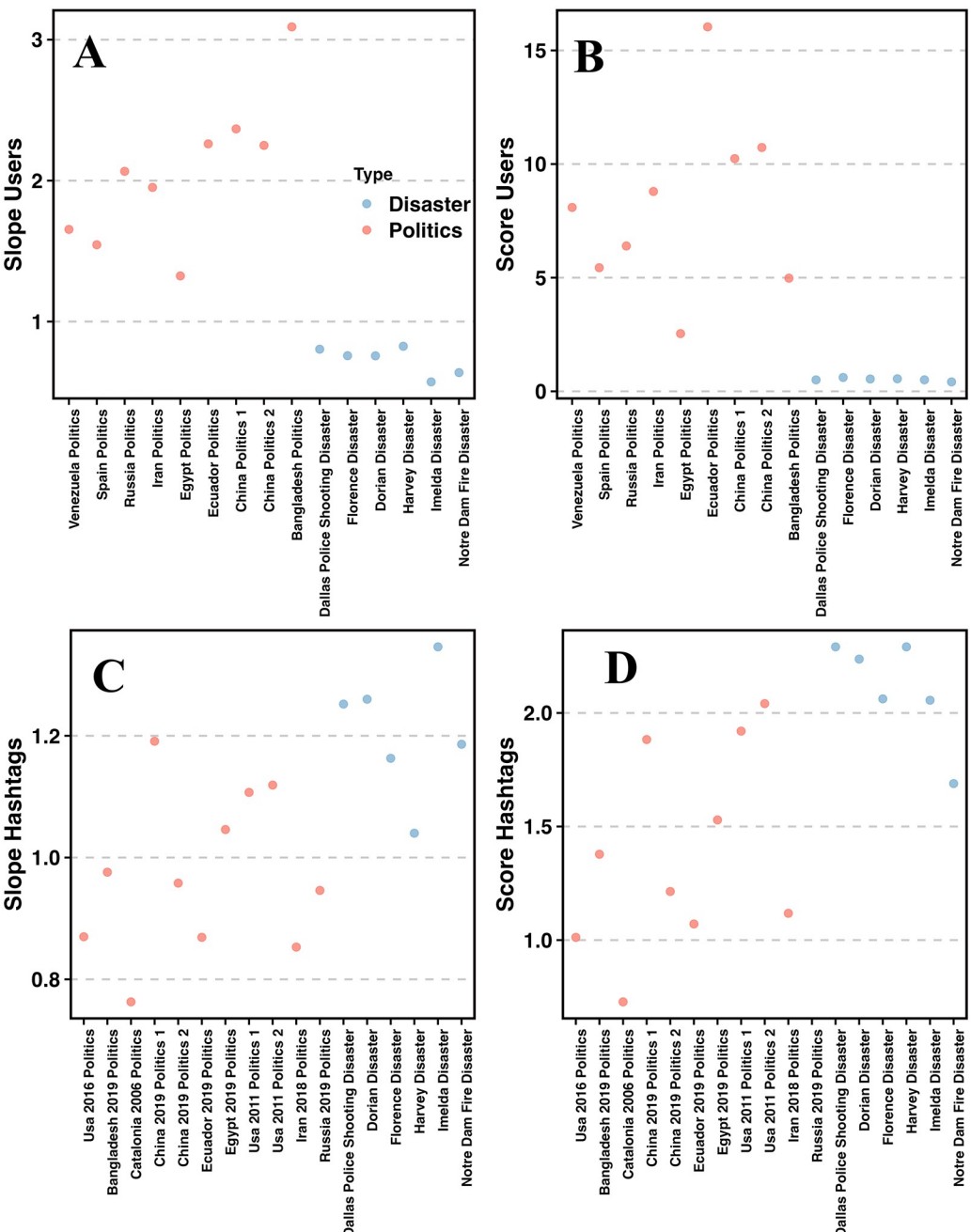

**Fig 4. Separation between political (orange) and disasters (blue) cascades.** (A) slope of user distribution, (B) PLMSE scores of user distribution, (C) slope of hashtags distribution, (D) PLMSE scores of hashtags distribution. Note the slope difference for the users and the hashtags and the differences between the political and disaster cascades. Means of political and non-political cascades are found significantly different. For the users distribution, the mean slopes are -2.05 vs. -0.59, (t-test p-value = 1.4E-05). Similarly, the user's PLMSE difference is 8.14 vs. 0.53 (t-test; p-value = 0.00045). For the hashtags, the mean slopes are -0.97 vs. -1.21 for the politics/disasters (t-test; p-value = 0.0018), while the PLMSE scores differences are 1.39 vs. 2.1 (p-value 0.0026).

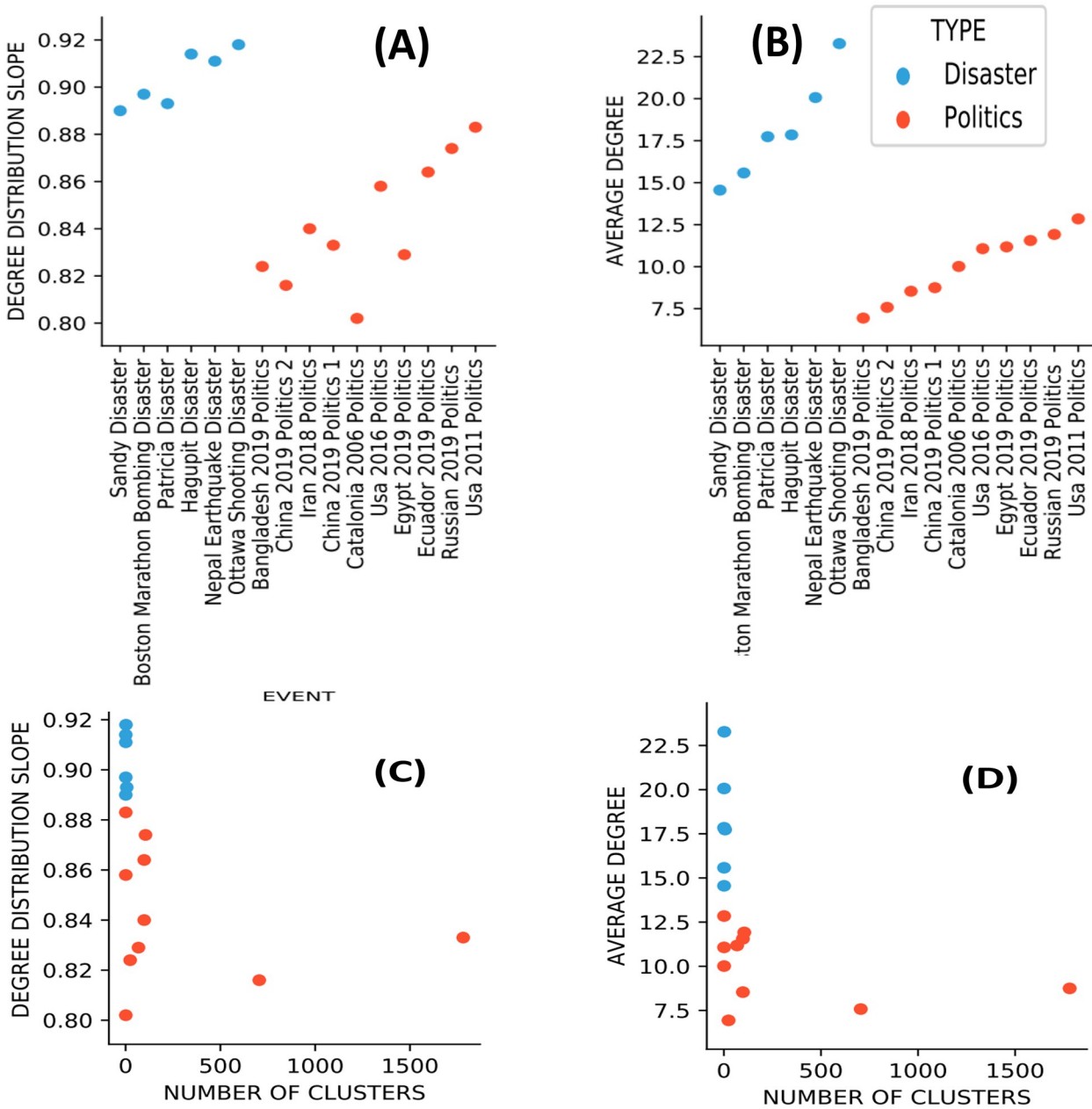

**Fig 5. Separation of political and natural events by word-to-tweet bipartite graphs.** (A)Slope of the degree distributions of the bipartite graphs. (B) The average degree of the bipartite graphs. (C) Two-dimension separation. A number of clusters in the graph—x-axis, degree distribution—y-axis. (D) Two-dimension separation. A number of clusters in the graph—x-axis, average degree—y-axis.

datasets (JSON format) from [52] where the complete tweet record of the disaster cascades was found. The political tweets were collected as before from Kaggle [53]. As before, the political topics appear in orange and the disasters appear in blue. In Fig 5(A) we show the slopes of the degree distribution for both types of cascades, and in Fig 5(B) we show the network's average degrees. Again, an interpretation of these results might suffer from our own confirmation bias and the data transformation makes the interpretation of these results sometimes less clear. Nevertheless, the separation is rather clear and can be observed in Fig 5(A)–5(D). Also, we see that

adding the number of clusters to the separation adds some information, but not such that deeply changes the separation by a single dimension.

## Discussion

We collected from Twitter information cascades of two different types—political topics and disasters. The disasters included topics such as earthquakes, hurricanes, mass shootings, and large fires. The comparison between these two types of cascades is based on our previous preliminary results [54], suggesting an intrinsic difference between the political and the non-political cascades. This difference is mainly due to an effort to influence the spread of political discussions and also due to the nature of political discussions to include a greater degree of repetitiveness.

In contrast, large-scale disasters generate more genuine social (media) discussions, which are based on an authentic human need to discuss, worry, and mostly share information in times of stress, or seek helpful information in times of disaster. Political campaigns, on the other hand, try to use social networks for their benefit, as a tool to draw attention, influence potential voters, or debate with political opponents. The goals and internal motivations of users who tweet about politics and those who tweet about large-scale disasters might also be different to some degree, and in this work, we try to capture these differences on a statistical macro scale level.

We find that the distribution of users in the political cascades strongly deviates from the power law distribution. We cannot truly claim that in the political cascades, the distribution of users follows a power law at all. The sharp drop in the lower right side of all the political cascades distribution (as observed in Fig 3), indicates a different user's distribution in politics compared to the natural disasters.

As for the hashtag distributions, we see in the political topics a slower decay (slope) indicating more hashtags that are used repetitively.

We develop also here a quantification method that detects the deviation from the natural power-law line. We name this measure the PLMSE in Eq 1, then demonstrate its efficiency. One should note that future directions can improve the PLMSE equations where one such possible candidate is $PLMSE = \frac{1-\psi}{\psi} \cdot \sum_{i=1}^{n} \psi^i \cdot \log^2\left(\frac{y_i}{C} \cdot i^\alpha\right)$, where the tuning parameter $0 \leq \psi \leqslant 1$, should be searched to find its optimal value. While we do not claim Eq (1) is optimal, by demonstrating its efficiency, we open the path to further research in the direction of measuring the fit to a power law by a mean square errors (PLMSE) formula, between the theoretical line and the data.

Another issue and possible limitation that needs to be considered is the fit of the distribution to the power law distribution. Some studies claim that the distribution of hashtags is indeed a power law [55], while others claim it is some generalized Zip's law [56]. While a true fit of many distributions to the power law is more seldom than expected [57], such an exact fit is not critical in the practical aspect of our work. This is because the conceptual novelty of our work is the ability to differentiate between political and non-political cascades based on statistical patterns of repetitions or words and users. These patterns of repetitions are even correct if the distribution does not fit an exact power law distribution but simply to any fat-tailed distribution.

We used logarithmic binning and least square methods after binning on a logarithmic scale. While some researchers claim that Maximum Likelihood (ML) method, that is strongly advocated by Clauset and others [57, 58], is the correct method to estimate the Power Law slope, other researchers, after generating power law distributions with known parameters and seeking their (known) slopes, claim otherwise [44]. This experimental work claims that "...*the*

*criticism about the inaccuracy of LSE in fitting power-law distributions is complete nonsense*", . . . and also that ". *Our experiments uncover a fundamental flaw in the widely known CSN2009 method proposed by Clauset et al, it tends to discard the majority of power-law data and fit the long-tailed noises.*

Furthermore, it has also been shown by [59], that in cases when the data is not a clean Power Law, but rather an approximation of the power law (as in our case), then a logarithmic binning will result in an unbiased slope estimate using LSE methods, and will perform better compared to the ML method. Along this line, we also tried using the method of Maximum Likelihood [57, 58], but it simply did not result in slopes that fitted our data visually, and since we could observe the slopes differences visually, we preferred to stick to slope measuring method by logarithmic binning and LSE that better describes the visual slope and it was observed in the date, and as it captures correctly the viewed differences between the political and non-political discussions.

A limitation that needs to be also discussed is that we used both the tweets and the retweets, which is not directly information cascade. Also, since in twitter, if user B retweets user A, and user C retweets user B, the data will appear as if user B and C retweeted user A, and the real path will be kept hidden. In this sense, the cascades in our data set do not truly represent direct cascades, but rather discussions on topics. For example, if a hashtag #Harvey appears after the hurricane Harvey, then we collect the discussion related to the hurricane, which was not discussed before the hurricane occurred. While the tweets were collected in the time where the hashtag was relevant, we have no true knowledge of the exact cascade structure in terms of who spread who. We thus in fact measure the discussion about a topic, and not necessarily the direct cascade spread from one person to another.

Another issue that needs to be considered is related to the users' distributions. While the distribution of users is highly different between the political and the nonpolitical cascades, as seen clearly in Fig 3 we are also more careful about counting on this difference. One should note that Twitter allows regular (noncommercial) users a limit of 5000 accounts, but paid accounts might not have this limitation. We are not sure what is the attitude of X toward this issue. Thus, the extreme difference between the user's and the disaster's cascades may be also due to political cascades including more paid accounts.

Last, interesting future direction can inspect additional topics, such as sports, fashion, and music hits, to see if these behave like the political or the disaster cascades. An example of such a topic is the Covid-19 pandemic, which created a global movement of vaccine supporters and deniers. A question arises if COVID-19 vaccine-related discussions were more similar to political or disaster cascades. Based on a single cascade data set from [60], we can see (S2 Fig) that the word distributions of this COVID-19 cascade have a slope of ($slope = -0.648$, $R^2 = 0.96$), which better resembles a political than a disaster cascade. This only draws a possible further study to determine if COVID-19-related information was politicized.

Also, Indeed, the line separating between propaganda, information bias and simple repetitions in twitter is a fine line. First, we will discuss the relations between information bias and propaganda, then discuss the role of authentic users in the spread of propaganda. We name in this work an information bias by any external force, aiming to bias the natural spread of ideas, as propaganda. This is a very broad definition, since it also includes commercial communication efforts to spread a name of a brand, as well as the messages from a charismatic opinion leader. This definition also includes the "darker" aspects of this phenomena such as the use of centrally or decentralized organized bots and fake accounts to spread misinformation. What we see in our work is that although the existence of charismatic opinion leaders that repeat consistently some specific messages is for sure a natural phenomenon, (and cannot be considered as propaganda by itself), the distribution of such charismatic users and topics is more

common in the political topics compared to the natural disaster's topics. This can be clearly observed in Figs 2 and 3, where one can observe that the distribution slopes for the hashtags used in the political topics are less steep, thus, the decay in hashtags usage is slower in the political topics, representing higher repetitiveness in the language of the political topics. In contrast, for the users, the slope is steeper for politics, thus some users are very active while the majority are barely active at all. These active users in the politics, that repeatedly spread the same hashtags again and again, and are thus the statistical footprint of propaganda. Furthermore, as long as we keep the neutral definition of propaganda, as a repetitive broadcast of a message, regardless if this is an acceptable or unacceptable message, we avoid the need for any evaluation of the good or bad moral aspect of the topic. In this perspective, "Just do it" is a propaganda similarly to "From the river to the sea, Palestine will be free", since we ignore our personal view of the content, and just inspect if the distribution of these topics is biased or unbiased. Indeed, in general the term "propaganda" is used for a "bad" spread of ideas that includes a massive brainwash. We claim that any information bias is considered as propaganda, either if it's for good or for bad reasons. Furthermore, while the reviewer's observation that "yes we can" can be a legitimate message that has been spread by an authentic user, our "statistical" definitions of propaganda does not limit the term propaganda only to the direct effect between the government and the people. This is because such a narrow definition, will not define the burning of old scripts (either by German Nazy or by Young Mao enthusiasts), when no government gave a direct order to burn such books, as a natural phenomenon, but clearly as an act resulting of propaganda, even if such acts were not directly including the government, but only resulted in a crowd-to-crowd direct messages spread. To conclude: in this work we define propaganda as any effort to bias the spread of information. This effort is generally hidden. Bots, fake accounts, smart slogans, are all a part or this effort, in which although most effects cannot be observed individually, our claim here is that they can be observed by their statistical properties.

Last, with the fast emergence of large language models, such as GPT-4 or Google Gemini, we expect a greater sophistication in machine-related content generation and smart bots. Our method can be more valuable than before considering this coming future.

## Conclusion

We inspect several information cascades, of which one group of cascades is related to political topics while the other is to disasters, comprising in total of over 9 million records and 9 GB of data. We found that the initial distributions of hashtag repetitions and user repetitions differ ($p - value = 1.15E-5$) for users and ($p - value = 0.004$) for hashtags. These results suggest higher repetitiveness of hashtags in the political cascades which is manifested by a smaller exponent of the hashtag distribution compared to the disasters (non-political) cascades. As for the distribution of users, the political cascades in contrast have larger slopes, indicating a faster decay in user repetition, resulting from a tendency of one to either talk politics or barely discuss it at all. We additionally propose a quantification measure named PLMSE, which is based on the known MSE measure with adaptation to power law curves, and show how its use helps to separate the two types of cascades.

Overall, the distribution of users in the political discussions tends to generate larger slopes of $\alpha > 1.1$, while disasters tend to result in milder slopes of $\alpha < 0.67$ (Fig 4A). In the hashtag distribution, the political topic tends to generate smaller slopes with mean($\alpha$)=0.97, while the disasters hashtag distribution tends to result in a faster decay, and slopes with an exponent mean($\alpha$)=1.21 (see Fig 4C). As for the PLMSE scores, for the hashtags, their means values is 1.4 compared to the disasters where the mean score is 2.1. The user's mean score is 8.97

compared to the disaster's mean score which is 0.522. These differences are observed both visually as well as numerically in Fig 4B. When transforming the data into a word-to-tweet bipartite graph results in an additional separation space, and thus an even better separation between these two types of cascades.

We proposed here several attributes to estimate and separate between political and non-political cascades. It is commonly believed that the level of bias in the political cascades is higher, although we know that disaster cascades also include some degree of manipulation. Authentic cascades and discussions are assumed to be these discussions where people truly share and seek information, and are less likely to try to manipulate the flow of information. Political cascades, on the other hand, have a more defined goal—to convince others. We believe that our proposed method can be useful as a first high-level assessment tool to detect the presence of propaganda or bias in information cascades and discussions in social media. With the coming age of computational propaganda based on large language models tool, the ability to detect intentional bias, and an external effort to spread ideas, might more relevant than ever before.

## Supporting information

**S1 Table. Data sets used for hashtag analysis.**
(XLSX)

**S2 Table. Data sets used for user analysis.**
(XLSX)

**S3 Table. Data sets used for bipartite graphs network analysis.**
(XLSX)

**S4 Table. Hashtags data points.**
(XLSX)

**S5 Table. Users data points.**
(XLSX)

**S6 Table. Bipartite network data points.**
(XLSX)

**S1 Fig. Power law slope of COVID-19 bigrams (Banda et al, 2020).** The slope is more similar to the political cascades than it is to the disaster cascades.
(TIFF)

**S2 Fig. Additional slopes images (complementary to Figs 2 and 3 for hashtags of Egypt and Iran politics.**
(TIFF)

## Author Contributions

**Conceptualization:** Alon Sela, John Phillip Sabou, Miloš Ulman.

**Data curation:** Alon Sela, Omer Neter, Václav Lohr, Petr Cihelka, Fan Wang.

**Formal analysis:** Alon Sela, Omer Neter.

**Funding acquisition:** Alon Sela, Moti Zwilling, Miloš Ulman.

**Investigation:** Alon Sela.

**Methodology:** Alon Sela, Václav Lohr, Miloš Ulman.

**Project administration:** Alon Sela, Miloš Ulman.

**Resources:** Petr Cihelka, Miloš Ulman.

**Software:** Omer Neter, Petr Cihelka, Fan Wang.

**Supervision:** Alon Sela, Miloš Ulman.

**Validation:** Alon Sela.

**Visualization:** Alon Sela.

**Writing – original draft:** Alon Sela, Moti Zwilling, John Phillip Sabou, Miloš Ulman.

**Writing – review & editing:** Alon Sela, John Phillip Sabou, Miloš Ulman.

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
