## [Decision Letter · Decision Letter 0]

12 Sep 2023

PONE-D-23-25017“Signals of Propaganda” - Detecting and Estimating Political Influences in Information Cascades in Social NetworksPLOS ONE

Dear Dr. Sela,

Thank you for submitting your manuscript to PLOS ONE. After careful consideration, we feel that it has merit but does not fully meet PLOS ONE’s publication criteria as it currently stands. Therefore, we invite you to submit a revised version of the manuscript that addresses the points raised during the review process.

We look forward to receiving your revised manuscript.

Kind regards,

Gilad Ravid, Ph.D.

Academic Editor

PLOS ONE

Journal Requirements:

2. In your Methods section, please include additional information about your dataset and ensure that you have included a statement specifying whether the collection and analysis method complied with the terms and conditions for the source of the data.

4. Please note that funding information should not appear in any section or other areas of your manuscript. We will only publish funding information present in the Funding Statement section of the online submission form. Please remove any funding-related text from the manuscript.

6. Thank you for stating the following financial disclosure: 

   "no"

7. In your Data Availability statement, you have not specified where the minimal data set underlying the results described in your manuscript can be found. PLOS defines a study's minimal data set as the underlying data used to reach the conclusions drawn in the manuscript and any additional data required to replicate the reported study findings in their entirety. All PLOS journals require that the minimal data set be made fully available. For more information about our data policy, please see http://journals.plos.org/plosone/s/data-availability.

8. We note that you have stated that you will provide repository information for your data at acceptance. Should your manuscript be accepted for publication, we will hold it until you provide the relevant accession numbers or DOIs necessary to access your data. If you wish to make changes to your Data Availability statement, please describe these changes in your cover letter and we will update your Data Availability statement to reflect the information you provide.

Additional Editor Comments:

Please look at the detailed review by the reviewers. In addition to their helpful response, I would like to add the following review:

a. Propaganda is based on "partial truth or outright fiction" (page 2). Since you did not analyze the tweets' content, you cannot classify the tweets as true or not; hence, you actually measure and analyze the cascading of information. I think you should consider re-framing the paper as cascading rather than Propaganda

b. The example of tree height as power-law distribution should be mentioned in the citations you cite. The citations research the connection between tree diameter and its height. Actually, Barabasi and Albert discussed that height cannot be power-law distributed, as with such a case, we anticipate finding trees with 1 km of height.

c. Methodology – Please describe the criterion for collecting the data for each dataset. If data is collected by their hashtag, that might explain the PL distribution.

d. How is the data binned into bins?

e. PLMSE – mean squared error cost function can be implemented to the power law without modification (the mean of the square of the difference between the real value and the predicted one). What are the needs for the new equation? How is it related to the original concept?

f. On page 7, the authors describe the network analysis they conducted. The results of this analysis should have been reported—for example, betweenness centrality, GC, and more.

g. You mention the outlined frequency of the name of the disease hashtag (page 8). Since regression is sensitive to outliners, how will the results be changed if you omit this point?

h. Did you check the t-test assumptions? What flavor of t-test has been used? If you use two groups unequal variance test, the p-value significance is marginally

i. The data series for hashtags and users and the network data series differ. It seems like a cherry-picking series. It is better to combine all the datasets into one set so that you can perform all your research on them.

j. Have you considered that the politics users distribution is not a power law?

k. Fig 4 – What is the x-axis in graphs A and B? In all four graphs, there are more than six red points ( which are described in the SI tables)

l. Fig 5 The analysis on parts A and B does not need network analysis; they are the distribution of hashtags in tweets and tweets to hashtags, and B is the average number of hashtags.

Reviewers' comments:

Reviewer's Responses to Questions

**Comments to the Author**

1. Is the manuscript technically sound, and do the data support the conclusions?

Reviewer #1: Partly

Reviewer #2: Partly

2. Has the statistical analysis been performed appropriately and rigorously? 

Reviewer #1: Yes

Reviewer #2: I Don't Know

3. Have the authors made all data underlying the findings in their manuscript fully available?

Reviewer #1: Yes

Reviewer #2: Yes

4. Is the manuscript presented in an intelligible fashion and written in standard English?

Reviewer #1: Yes

Reviewer #2: Yes

5. Review Comments to the Author

Reviewer #1: Sela et al. explores cascades of tweets relating to political topics and disasters, finding that for disasters the cascade distribution better follows a power-law whereas for political topics, the distribution falls off more quickly. The authors hypothesize about why this might be, noting various ideas from the literature. The paper is on an interesting topic and suggests an interesting comparison, along with novel findings. At the same time, several points should be clarified before publication.

On page 2 the authors suggest they use the disaster information as a control group for political cascades. I’m a bit concerned about this as each group has unique aspects and there are not other types of information explored beyond disasters. Perhaps suggesting this is just a comparison rather than a control is sufficient, or more carefully qualifying the idea of this as a control group. Likewise, the comment on page 3 that disaster cascades are not likely to be manipulated is not supported and I am not sure that this is certainly true.

The PLMSE measure is not fully clear to me. Conceptually I believe the authors are fitting a power-law to the data and then asking how far from the fit the data appears. However, Eq. 1 does not really appear similar to a MSE of a fit. Perhaps they could show the derivation more carefully (in SI) to explain better? Likewise, the authors describe ’n’ as being the number of points, yet it seems to me that this might be the number of bins in the power-law distribution? Otherwise, how do they have an expectation for what each point should be (since they’re drawn from a distribution)?

In Figure 3, the authors don’t show the US elections it seems. Is there a reason for this? Are the results similar for the US as the others? Likewise the ‘user_index’ label is unclear to me. Is this just the number of users that e.g., have 1 tweet, 10 tweets, etc. on the topic? The caption for this and Fig. 2 should be more detailed.

The authors claim that the separation in Fig. 4 is clear between disasters and politics seems a bit overstated. Perhaps the authors could provide a metric on how good their method is at making predictions? Could they maybe calculate an AUROC based on the scores or give another standard metric? Eyeballing, the graph for example in 4D many points in politics are comparable to disasters (e.g., USA politics 2011).

The paper could use a thorough proofreading. For example, on page 1 ‘includes a repetitive broadcasting’ should be ‘includes repetitive broadcasting’; on page 2 ‘of a political discussions’ should be ‘of political discussions’; on page 5, ‘would be explain’ should be ‘will be explained’; also on page 5 ‘the researcher study’ should be ‘the researcher studies’; also on page 5 ‘get Power law’ should be ‘get a power law’; on page 6 ‘data sets where used’ should be ‘data sets were used’; also on page 6 ‘scarping’ should be ‘scraping’ and ‘scraps’ should be ‘scrapes’; on page 8 ‘twits’ should be ‘tweets’; on page 14 ‘additianl’ should be ‘additional’;

On page 2 the statement that bots are illegal might be overstated, rather I think it is just a violation of terms of service potentially.

Reviewer #2: The paper proposes the analysis of several datasets of Twitter posts to evaluate the statistical differences between political and non-political discussions.

The proposed evaluation starts from the hypothesis that political discussions (as propaganda) generally leverage the repetition of messages and involve a limited number of users when compared with non-political discussions. The authors are able to distinguish between political and non-political datasets by studying the properties of graphs generated with the different datasets. Moreover, the authors propose the construction of a bipartite graph in which each tweet is connected with the words it contains.

The results emerge by studying how closely the graph properties exhibit theoretical power-law distributions and focusing on the evaluated parameters to derive the slope and an additional score called PLMSE (power-law mean square error).

Strong points:

- an attractive and reasonable approach

- a pleasant writing style

Weak points:

- not convincing experiments

- lacking of crucial details

- limited number of tests

- detection motivation not fully plausible

- typos

The bipartite graph-based approach is attractive and sounds relevant with respect to the traditional NLP-based ones.

However, while the paper's approach is attractive, its overall quality and impact seem limited, and the results appear not entirely sound.

First of all, many important details appear to be overlooked when describing the evaluation methodology. For example, it is unclear how the authors processed the tweets and which words were included in the graphs (what about the stopwords and alike?) or which are the details of the operation "The data was binned prior to the slope estimation". The plots reported in the figures are inconsistent and incomplete: it is unclear why some datasets are used for showing some properties but not for others (for example, USA-2016 and Catalonia-2006 are missing in Figure 3).

Another questionable point is the claim on page 7, where considering "words used in the tweet regardless if they are hashtags or not" would remove the bias that "political users might tend to use hashtags more often than regular users". This way, political discussions will include the words related to the tweet *and* the hashtags downgraded to regular words.

The use of disaster-related datasets is a reasonable sample. However, it is only one of the possible types of "cascades" found on Twitter. How do the results differ when considering discussions about other events, like the Super Bowl, lengthy expected movies, or World Cup exhibitions?

Furthermore, motivating the proposed methodology to detect political discussion sounds relatively trivial, considering that the presence of given hashtags proves that a tweet is political.

Finally, several typos undermine the overall quality of the paper: it seems that the paper did not receive a last text revision. A prominent example is the word "sty;e" on page 11, showing that a spell-check was missing before the submission to a prominent venue like PONE.

6. PLOS authors have the option to publish the peer review history of their article (what does this mean?). If published, this will include your full peer review and any attached files.

Reviewer #1: No

Reviewer #2: No

---

## [Author Response · Author response to Decision Letter 0]

5 May 2024

Respond to editor comments and to each of the reviewers can be found in the attached "Respond to Reviewers- file.

Response to reviewers 

Thank you for the time and effort to read and comment our manuscript "Signals of Propaganda: Detecting and Estimating Political Influences in Information Cascades in Social Networks". Thank you also for your beneficial comments, which we feel truly helped us improve our manuscript.

In our study, we evaluate how information cascades that include a non-organic spread, i.e., which is suspected to include some bias resulting from an effort to spread an agenda, and as such named “computational propaganda”, can be detected on Twitter via their macro-scale network properties, without examining the text itself. 

Our proposed detection method is based on the property of rhetorical speech (which is a part of propaganda) to include a certain degree of repetition in the messages i.e., “brainwash” or “slogans”, and in the tendency of political discussions to repeat broadcasting similar messages to their target audience. 

This creates a distribution where several terms repeat more times than in an "organic" discussion, for example, we use large scale disasters as a comparison group. 

We show how such repetitions can be detected through their Power law exponents features, for both users and hashtags distribution. We show also how transforming of the cascades into bipartite graphs, connecting tweets and the words appearing in these tweets, can help find this type of repetitiveness when hashtags are lacking or in cases where hashtags are unevenly distributed between the political cascades and the comparison group. 

As a comparison group to the political information cascades, we use information cascades related to large-scale disasters. We believe these cascades reflect a more “organic” cascade type since people in harsh situations share their knowledge and emotions for the sake of pure human cause, and not for the sake of spread or personal gain, and thus disasters cascades are less centralized in their nature. 

Also, we developed a distinctive quantitive measure, named the PLMSE measure, based on our distinction above. We find that the political discussions tend to be more centralized, i.e., fewer users repeat similar words more often than it would have occurred in a n “organic” discussion. On the other hand, authentic social network conversations are more heterogeneous in terms of their used vocabulary. 

Following the reviewer`s comments, we added an entire section explaining our PLMSE formula. We also added statistical tests to show the separation`s significance as long as many other corrections. We reorganized the GitHub repository entirely to enable a reproduceable work.

Thank you for this opportunity and your consideration of this manuscript as well as for the additional time that we received for its corrections and the deep remarks which improved the readability and clarity of our work. 

Responses to the first reviewer are shown directly after this letter, while responds to the second reviewer starting at page 6. Additional comments and corrections following the editor's comments (which in some cases were similar to the reviewer`s comments) are allocated to the letter to the editor document and here starting at page 10. When the editor and the reviewer`s comments were on the same topic, we only answered the editor for to reduce redundancy.

We look forward to hearing back from you soon. 

With kind regards,

Alon Sela (on behalf of the authors). 

Reviewer 1 

Comment: Reviewer #1: Sela et al. explores cascades of tweets relating to political topics and disasters, finding that for disasters the cascade distribution better follows a power-law whereas for political topics, the distribution falls off more quickly. The authors hypothesize about why this might be, noting various ideas from the literature. The paper is on an interesting topic and suggests an interesting comparison, along with novel findings. 

At the same time, several points should be clarified before publication.

Answer: Thank you for this positive general evaluation.

Comment: On page 2 the authors suggest they use the disaster information as a control group for political cascades. I’m a bit concerned about this as each group has unique aspects and there are not other types of information explored beyond disasters. Perhaps suggesting this is just a comparison rather than a control is sufficient, or more carefully qualifying the idea of this as a control group. Likewise, the comment on page 3 that disaster cascades are not likely to be manipulated is not supported and I am not sure that this is certainly true.

Answer: Thank you for this comment. Indeed, following the reviewer’s comments, we expressed our belief of a weaker degree of manipulation in disasters compared to politics in a more careful way. 

Comment: The PLMSE measure is not fully clear to me. Conceptually I believe the authors are fitting a power-law to the data and then asking how far from the fit the data appears. However, Eq. 1 does not really appear like a MSE of a fit. Perhaps they could show the derivation more carefully (in SI) to explain better? Likewise, the authors describe ’n’ as being the number of points, yet it seems to me that this might be the number of bins in the power-law distribution? Otherwise, how do they have an expectation for what each point should be (since they’re drawn from a distribution)?

Answer: Thank you for this important comment. We added to the article an explanation of the PLMSE formula. Indeed, the PLMSE formula was not clear, and we hope now it is clearer. 

Comment: In Figure 3, the authors don’t show the US elections it seems. Is there a reason for this? Are the results similar for the US as the others? Likewise, the ‘user_index’ label is unclear to me. Is this just the number of users that e.g., have 1 tweet, 10 tweets, etc. on the topic?

Answer: Thank you for this comment. In figure 3 we show 6 vs 6 politics and disasters, and in fig 4 we show the entire data. We simply showed part of the data as histograms due to space limits. We added to the git the additional images from which we compute the slopes which are the values that separate between the numbers. As for the user_index, indeed it is the number of user when sorted from the most to the least frequently appearing user in the data. 

Comment: The caption for this and Fig. 2 should be more detailed.

Answer: Thank you for this comment. We added to the caption the following explanation.

Comments: The authors claim that the separation in Fig. 4 is clear between disasters and politics seems a bit overstated. Perhaps the authors could provide a metric on how good their method is at making predictions? Could they maybe calculate an AUROC based on the scores or give another standard metric? Eyeballing, the graph for example in 4D many points in politics are comparable to disasters (e.g., USA politics 2011).

Answers: Thank you for this important comment. We added a t-test and show the p-values which were computed and added to the manuscript in the caption of Fig.4. 

Comments: repetitive broadcasting’ should be ‘includes repetitive broadcasting’; on page 2 ‘of a political discussions’ should be ‘of political discussions’; on page 5, ‘would be explain’ should be ‘will be explained’; also on page 5 ‘the researcher study’ should be ‘the researcher studies’; also on page 5 ‘get Power law’ should be ‘get a power law’; on page 6 ‘data sets where used’ should be ‘data sets were used’; also on page 6 ‘scarping’ should be ‘scraping’ and ‘scraps’ should be ‘scrapes’; on page 8 ‘twits’ should be ‘tweets’; on page 14 ‘additianl’ should be ‘additional’.

Answer: Thank you and our apologies for this issue. We thoroughly spell check the article again. We believe that now all spelling mistakes have been corrected. 

Comments: On page 2 the statement that bots are illegal might be overstated, rather I think it is just a violation of terms of service potentially.

Answer: Thank you for this comment, we rewrote the sentence in a more subtle manner to correctly describe the reality where bots in many cases are legal (see below):

Reviewer 2 

Comment: The paper proposes the analysis of several datasets of Twitter posts to evaluate the statistical differences between political and non-political discussions.

The proposed evaluation starts from the hypothesis that political discussions (as propaganda) generally leverage the repetition of messages and involve a limited number of users when compared with non-political discussions. The authors are able to distinguish between political and non-political datasets by studying the properties of graphs generated with the different datasets. Moreover, the authors propose the construction of a bipartite graph in which each tweet is connected with the words it contains.

The results emerge by studying how closely the graph properties exhibit theoretical power-law distributions and focusing on the evaluated parameters to derive the slope and an additional score called PLMSE (power-law mean square error).

Strong points:

- an attractive and reasonable approach

- a pleasant writing style

Answer: Thank you very much for this positive evaluation. We will do our best to address the weak points. 

Comment: 

Weak points:

- not convincing experiments

- lacking of crucial details

- limited number of tests

- detection motivation not fully plausible

- typos

Answer: Thank you for this evaluation. We added t-tests to statistically support our claim that the political and nonpolitical cascades differ. We did not perform experiments, but rather analyzed large datasets where some include millions of tweets. All together, we analyzed networks with over 19 million edges and 3 million nodes. For the analysis that did not include the full cascade network, we analyze almost 24 million records, of which we concluded these results.

This is not a negligible data set, and each cascade topic can be of many millions' records. The full data sets sizes and records used for the network analysis and non-network analysis are found in the Git of the project in a file named "DataSources.xlsx" and Network_datasets.csv". We corrected the comments and concerns raised by the reviewers. 

We added the statistical significance to estimate the difference between the attributes of the political /non-political cascades and added also a mathematical explanation to our PLMSE method of numerical estimation of the cascade`s type. 

Typos have been corrected. 

Comment: The bipartite graph-based approach is attractive and sounds relevant with respect to the traditional NLP-based ones. However, while the paper's approach is attractive, its overall quality and impact seem limited, and the results appear not entirely sound. 

First of all, many important details appear to be overlooked when describing the evaluation methodology. For example, it is unclear how the authors processed the tweets and which words were included in the graphs (what about the stop words and alike?) or which are the details of the operation.

Answer: Thank you for this comment. We added the following paragraph to the article to clearly explain this process.

Comment: "The data was binned prior to the slope estimation". The plots reported in the figures are inconsistent and incomplete: it is unclear why some datasets are used for showing some properties but not for others (for example, USA-2016 and Catalonia-2006 are missing in Figure 3).

Answer: Thank you for this comment. Your observation that “some datasets are used for showing some properties but not for others” in Figure 3 is correct. Figs 2 and 3 show each 6 data sets of political and 6 of nonpolitical cascades. Figs 4 and 5 in comparison, show 6 disaster cascades but 10 points (and not 6) related to political cascades. The reason why we show in Fig 2 and 3 only 6 X 2 images is simply for ease of presentation space and since we only found 6 disaster’s datasets of large enough size. 

Furthermore, based on figs. 2,3, one cannot truly conclude that these images are different. It is rather more of a demonstration that illustrates the phenomena and the deviation, mainly on the upper left side in the political cascades. The images show the conceptual idea but does not quantify it. This quantification is shown in fig 4 and 5 in contrast, where each cascade is only marked by a single point (and not as an entire graph). Thus, we can show all data sets that were available to us, which are 10 political cascades and 6 disaster cascades. 

Following the reviewer`s comment, we corrected the caption of images 2 and 3 and emphasized that this is only a part of the datasets. 

In addition, we add to the SI section a table containing the data points for all 10 political points and 6 disaster points. Following the reviewer's comments we added this paragraph

Comment: Another questionable point is the claim on page 7, where considering "words used in the tweet regardless if they are hashtags or not" would remove the bias that "political users might tend to use hashtags more often than regular users". This way, political discussions will include the words related to the tweet *and* the hashtags downgraded to regular words.

Answer: Thank you for this correct comment. When analyzing the tweets, we removed first stop words. We performed a standard lemmatization which include this work removal. Thus, we do not include in our analysis of bipartite graphs words such as and / or / etc. in the tweet. We used the NLTQ and scikit-learn python packages that do this already. 

Comment: The use of disaster-related datasets is a reasonable sample. However, it is only one of the possible types of "cascades" found on Twitter. How do the results differ when considering discussions about other events, like the Super Bowl, lengthy expected movies, or World Cup exhibitions?

Answer: Thank you for this comment. We understand the point of comparing additional cascades to the 16 available cascades. We added the Covid19 frequent bigrams cascades to the SI section, and saw that according to its slope, Covid 19 cascades are more similar to a political issue than it resembles a disaster. This is since its slope is -0.648. We plan to perform such an analysis on the coming USA election, but this analysis is out of the scope of the current work.

Comment: Furthermore, motivating the proposed methodology to detect political discussion sounds relatively trivial, considering that the presence of given hashtags proves that a tweet is political.

Answer: Thank you for this comment. Truly finding political topics is trivial. Our method does not offer only method to find pollical discussions (which clearly does not much effort), but rather finds interventions in a discussion that are not “organic” in the sense that the distributions deviates from the organic discussion. We use politics since we know that politics has a clear goal – to push an agenda. But in other cases, and topics this is not clear. For example, in health-related topics, it is not always clear if it’s an organic discussion or not. Furthermore, if we do not know in advance if a discussion is political or not, for example Global Warming or Immigration, our method permits us to determine for any given topic if its more organic (authentic users discussion) or more politic (specific key holder that aim at spreading their agenda. One should note that we did not inspect the textual meaning or content of the hashtag itself, nor if it was a hashtag or a simple word. We only look at the distributions and their slopes parameters. 

The method used does not pretend to distinguish solely political propaganda, but mainly an amplification of some words, which appears in political messages but also might appear in commercial communication. 

Indeed, we mention in the introduction on page 2 that “Pre-election political messages tend to be more aggressive, populistic and in general, use similar techniques as the ones used in commercial communication. These commercial communication techniques repetitively broadcast a few well-defi

---

## [Decision Letter · Decision Letter 1]

26 Jul 2024

PONE-D-23-25017R1Signals of Propaganda - Detecting and Estimating Political Influences in Information Cascades in Social NetworksPLOS ONE

Dear Dr. Sela,

Thank you for submitting your manuscript to PLOS ONE. After careful consideration, we feel that it has merit but does not fully meet PLOS ONE’s publication criteria as it currently stands. Therefore, we invite you to submit a revised version of the manuscript that addresses the points raised during the review process.

We look forward to receiving your revised manuscript.

Kind regards,

Gilad Ravid, Ph.D.

Academic Editor

PLOS ONE

Journal Requirements:

Additional Editor Comments:

The authors improved the paper and answered most of the reviewers' concerns. Reviewer 1 is satisfied with the results, but Reviewer 2 did not respond to requests to review the paper again.

Some minor issues still need to be resolved before we publish the paper.

a. the paper positions itself as dealing with propaganda and cascade. As they appear in the paper, the two phrases are not used as their true meaning. As Twitter is a crowded, uncontrolled system, it is not precise to name information distribution as propaganda (neither good nor bad). This is even stronger when the authors examine hashtags, which, as they claim, are keywords that describe the tweet. Hence, if many use the same hashtag, it is not a sign of propaganda. For example, many use the hashtag "#YesWeCan" to indicate their support for Obama's campaign, but it is hard to call it a sign of propaganda. Also, I need help finding a relationship between the users' power law distribution and propaganda. As regards the cascade term, the research does not look at the cascade of information; cascade happens when someone transfers ideas or information he hears from someone else. There is no indication of transferability in the study.

The authors use the term "political cascade" to denote the political networks. Political can cascade information, ideas, and knowledge but can't cascade politics.

Line 242: should be a curve fit line (you miss the "v").

Line 252 vs. line 267: Yi is index or frequency (I think the letter).

Line 273: Please use the standard notation in the equation; the predicted value is Y hat (^), and the upper bar is reserved for the mean.

Line 280: both sides of the equations are exactly the same.

Throughout the article and equations, the authors interchangeably treat alpha as slop and alpha as minus the slop. For example, line 333 states that alpha equals -763 (you miss the decimal point. It should be -0.763), but on line 276 the line equation considers alpha as positive (the minus is part of the equation)

Please add comments and refer to some prior works on the differences and problems of fitting power law directly vs. fitting to the log-log of the values.Goldstein, M. L., Morris, S. A., & Yen, G. G. (2004). Problems with fitting to the power-law distribution. The European Physical Journal B-Condensed Matter and Complex Systems, 41, 255-258. is one candidate reference ( the problem related to the changing distribution of the log(error) you introduce)

line 254 vs. line 312: Did you make least square error fitting or maximum likelihood fitting?

Line 306: In your answers to the reviewers, you stated that you didn't perform betweenness calculations; in the paper, you did.

Line 337 vs. Fig 2, Bangladesh 2019 slope is 0.976

Line 340 vs. caption of Fig 2. What is the mean 0.97 or 0.938

Reviewers' comments:

Reviewer's Responses to Questions

**Comments to the Author**

1. If the authors have adequately addressed your comments raised in a previous round of review and you feel that this manuscript is now acceptable for publication, you may indicate that here to bypass the “Comments to the Author” section, enter your conflict of interest statement in the “Confidential to Editor” section, and submit your "Accept" recommendation.

Reviewer #1: All comments have been addressed

2. Is the manuscript technically sound, and do the data support the conclusions?

Reviewer #1: Yes

3. Has the statistical analysis been performed appropriately and rigorously? 

Reviewer #1: Yes

4. Have the authors made all data underlying the findings in their manuscript fully available?

Reviewer #1: Yes

5. Is the manuscript presented in an intelligible fashion and written in standard English?

Reviewer #1: (No Response)

6. Review Comments to the Author

Reviewer #1: The authors have addressed my concerns.

In particular they added a section on their method of PLMSE better explaining the model. They also added statistical tests that are more convincing for the segregation between the datasets.

7. PLOS authors have the option to publish the peer review history of their article (what does this mean?). If published, this will include your full peer review and any attached files.

Reviewer #1: No

---

## [Author Response · Author response to Decision Letter 1]

14 Aug 2024

A detailed response to reviewers' document has been uploaded to the system.

---

## [Editor Report · Decision Letter 2]

16 Aug 2024

Signals of Propaganda - Detecting and Estimating Political Influences in Information Cascades in Social Networks

PONE-D-23-25017R2

Dear Dr. Sela,

We’re pleased to inform you that your manuscript has been judged scientifically suitable for publication and will be formally accepted for publication once it meets all outstanding technical requirements.

Kind regards,

Gilad Ravid, Ph.D.

Academic Editor

PLOS ONE

Additional Editor Comments (optional):

All comments addressed appropriately
---

## [Editor Report · Acceptance letter]

7 Nov 2024

PONE-D-23-25017R2 

PLOS ONE

Dear Dr. Sela, 

I'm pleased to inform you that your manuscript has been deemed suitable for publication in PLOS ONE. Congratulations! Your manuscript is now being handed over to our production team.

Kind regards, 

on behalf of

Prof. Gilad Ravid 

Academic Editor

PLOS ONE